# Advanced Patch-Based Affine Motion Estimation for Dynamic Point Cloud Geometry Compression

**DOI:** 10.3390/s24103142

**Published:** 2024-05-15

**Authors:** Yiting Shao, Wei Gao, Shan Liu, Ge Li

**Affiliations:** 1School of Electronic and Computer Engineering, Peking University, Shenzhen 518055, China; ytshao@pku.edu.cn (Y.S.); geli@ece.pku.edu.cn (G.L.); 2Peng Cheng Laboratory, Shenzhen 518066, China; 3Media Lab, Tencent, Palo Alto, CA 94306-2028, USA; shanl@tencent.com

**Keywords:** dynamic point cloud geometry compression, affine motion estimation, patch generation

## Abstract

The substantial data volume within dynamic point clouds representing three-dimensional moving entities necessitates advancements in compression techniques. Motion estimation (ME) is crucial for reducing point cloud temporal redundancy. Standard block-based ME schemes, which typically utilize the previously decoded point clouds as inter-reference frames, often yield inaccurate and translation-only estimates for dynamic point clouds. To overcome this limitation, we propose an advanced patch-based affine ME scheme for dynamic point cloud geometry compression. Our approach employs a forward-backward jointing ME strategy, generating affine motion-compensated frames for improved inter-geometry references. Before the forward ME process, point cloud motion analysis is conducted on previous frames to perceive motion characteristics. Then, a point cloud is segmented into deformable patches based on geometry correlation and motion coherence. During the forward ME process, affine motion models are introduced to depict the deformable patch motions from the reference to the current frame. Later, affine motion-compensated frames are exploited in the backward ME process to obtain refined motions for better coding performance. Experimental results demonstrate the superiority of our proposed scheme, achieving an average 6.28% geometry bitrate gain over the inter codec anchor. Additional results also validate the effectiveness of key modules within the proposed ME scheme.

## 1. Introduction

With the development of three-dimensional (3D) sensor technology, significant strides in point cloud capturing and reconstruction have spurred a surge in interest in 3D media applications, including virtual reality, immersive telepresence, and free-viewpoint television [1,2,3]. Point clouds are widely adopted for representing both static and dynamic objects in 3D space. A point cloud comprises a sparse and unstructured collection of points enriched with position, color, and additional attributes like reflectance and transparency. However, the escalating demand for high-resolution and high-bit-depth point clouds presents a formidable challenge due to the substantial data volume they entail. This challenge is particularly pronounced for applications constrained by limited transmission bandwidth and storage capacity. Consequently, there is an urgent need for efficient compression techniques to mitigate the spatial and temporal redundancy inherent in point clouds.

Considerable research endeavors have been dedicated to static point cloud geometry compression [4,5,6], static point cloud attribute compression [7,8,9,10], and intra-codec design [11,12,13]. Despite these advancements in removing spatial information redundancy in point clouds, the domain of dynamic point cloud compression remains relatively unexplored. Dynamic point clouds inherently possess significant temporal redundancy attributed to moving people and objects, often exhibiting regular motion patterns. The primary challenge in dynamic point cloud compression lies in establishing inter-frame point correspondences within unstructured point sets and exploiting temporal correlation to generate a more compact point cloud representation [14,15,16]. It is noteworthy that point correspondences in point cloud sequences are implicit, and the number of points may vary across different frames. These characteristics pose challenges in reducing temporal redundancy during point cloud compression. Given that geometric fidelity profoundly influences point cloud quality, our focus in this paper is on dynamic point cloud geometry lossless compression.

Motion estimation (ME) has been proven to be effective in point cloud temporal redundancy removal [17,18,19]. In dynamic point clouds, numerous independently moving objects and components necessitate efficient approximation of the intricate motion field. The accuracy of ME significantly influences geometry inter-coding performance. A notable ME-based inter-coding framework is the inter-exploration model for geometry-based point cloud compression (G-PCC interEM) introduced by the Moving Picture Experts Group 3D Graphics coding group (MPEG 3DG) [20]. In G-PCC interEM [21], the previously decoded frame is directly employed as the inter-reference for the current frame being coded. Then, the current point cloud undergoes partitioning into regular geometry blocks via an octree, with the motion field represented as block-based translational motions. Block-matching motion search is conducted in the interEM to capture the motions of all blocks in dynamic point clouds.

Several critical issues in current ME schemes warrant attention. First, the selection of the reference frame in the interEM relies solely on the previously decoded frame without any warping operations. This would lead to inaccurate block-matching pairs in the backward ME process when it occurs in fast-moving areas. We intend to introduce a forward-backward jointing ME scheme to generate a better reference frame to enhance the accuracy of the ME process. Second, the octree-based block structure in the interEM limits motion representation to capturing discontinuities along three axes. This poses challenges in fine-tuning motion representation due to the fixed size of the block. To address this, we propose a motion-assisted patch generation approach for flexible motion representation, facilitating better alignment between the estimated motion representation with the real motion field in point clouds. Last but not least, the limitation of the translational motion model in the interEM restricts motion representation to three degrees of freedom (DOFs), hindering optimal inter-coding performance. We advocate for the incorporation of affine motion models to allow for accurate motion approximation with increased DOFs.

Overall, we propose an advanced patch-based affine ME framework incorporating a novel motion representation and estimation scheme for dynamic point cloud geometry compression. Our contributions can be summarized as follows:We design a forward-backward jointing ME strategy incorporating forward motion tracking and backward motion refinement. Forward motion tracking is conducted to generate better motion-compensated frames for improved inter-geometry references before the backward ME process. Results demonstrate that the proposed scheme notably improves the ME accuracy and coding performance.We propose a motion-assisted patch generation scheme for the flexible motion representation of point clouds. Motion priors from previously decoded frames are extracted to guide deformable patch generation. Our irregular patch representation can better depict the varying local motions in point clouds.We introduce an affine motion model to replace the traditional translational model to improve the ME accuracy for dynamic point clouds. The proposed affine motion model incorporates more DOFs, allowing for finer motion representation, thereby improving the precision of motion-compensated predictions and optimizing point cloud compression efficiency.

The paper is structured as follows. Section 2 provides a literature review on dynamic point cloud geometry compression and point cloud ME schemes. In Section 3, the details of the proposed framework are presented. Section 4 contains experimental results and analysis. Finally, the paper concludes in Section 5.

## 2. Related Work

### 2.1. Dynamic Point Cloud Compression

Current approaches to dynamic point cloud geometry compression can be categorized into two groups based on their utilization of ME methods: those without ME and those with ME. Point cloud inter codecs without ME typically disregard inter-frame movements and directly utilize previously decoded frames as inter references [22,23]. Conversely, inter coders with ME aim to identify optimal motions for temporal redundancy removal [21,24]. For instance, in the case of point cloud inter coders that do not utilize ME, early work by Kammerl et al. [25] employs the exclusive-OR (XOR) operation on geometry octree occupancy to reduce temporal redundancy among consecutive frames. Garcia et al. [22] propose a context-adaptive arithmetic coder without ME, utilizing a reference octree to construct temporal contexts. Milani et al. [26] introduce a transform-based coding approach employing multiple nonlinear context-related transforms tailored for dynamic point clouds. However, these approaches without ME may struggle to compress dynamic point clouds with large complex motions like [23].

A well-designed ME module significantly enhances coding performance. It is commonly addressed through methods such as the iterative closest points algorithm (ICP) [24,27], feature matching [28], and block matching [21,29]. Mekuria et al. [24] utilize ICP to estimate block-wise motions for point cloud inter coding. Thanou et al. [28] introduce a graph-based feature matching technique for motion estimation. However, the estimated motions in these approaches have not been well optimized in a R-D manner [24,28]. Santos et al. [29] propose an R-D optimized mode decision scheme in inter-frame prediction to improve coding performance. Additionally, Kim et al. [30] propose a skeleton-based nonrigid ME scheme for the compression of dynamic human point clouds, albeit with limited generality. Shao et al. [17] devise a registration-based ME scheme to capture nonrigid motions for different types of dynamic point clouds. Notably, the G-PCC interEM [21] achieves remarkable coding performance with an R-D optimized ME scheme for dynamic point clouds. However, limited updates to local ME have identified it as a bottleneck in codec design, prompting exploration into advanced local ME frameworks to enhance the coding performance for dynamic point clouds.

### 2.2. Point Cloud Motion Estimation

A precise local ME scheme is essential for maximizing the utilization of temporal correlation in dynamic point cloud compression. The design of ME schemes primarily revolves around two critical aspects: the motion field representation and the motion model. Regarding the motion field representation for point cloud moving contents, the most popular method is the block-based motion field derived from the block-matching strategy. G-PCC interEM [21] is a pioneer in block-matching ME for dynamic point cloud compression. Based on this, An et al. in [31] enhance the interEM with new block-matching criteria, leading to notable improvements in geometry bitrate gains and coding time efficiency. Moreover, Hong et al. in [15] propose a fractional-voxel ME method to accommodate the inherent distinctions between dynamic point clouds. This scheme specifically addresses the irregular distribution of point cloud geometry between consecutive frames. However, block-based motion field representation assumes uniform motion within each block, disregarding object boundaries and complex motion scenarios. To address this limitation, Thanou et al. in [28] introduce a graph-based motion field representation with a graph-matching ME scheme. Despite partially mitigating the shortcomings of block-based approaches, the graph construction and matching processes are computationally intensive.

Another key issue in the design of the ME schemes is the selection of motion models. In general, motion models employed in dynamic point clouds can be classified into four categories based on their complexity: translational, rigid, affine, and nonrigid models [17]. Translational motion models are frequently utilized in dynamic point clouds owing to their simplicity, exemplified by the block-based translational motion estimation in G-PCC interEM [21]. Additionally, translational motions are estimated in [28] through feature matching between successive graphs of point clouds. Nonetheless, this approach is time-consuming due to the graph transform-based spectral feature generation for points. A rigid motion model offers more DOFs in describing motion, encompassing rigid transformations in 3D space, such as rotations, translations, and reflections. Mekuria et al. in [24] adopt ICP to estimate rigid transformations among blocks in consecutive frames and each estimated transformation consists of a rotation matrix as a quaternion and a translation motion with three parameters. Moreover, an affine motion model for dynamic point clouds can represent a broader range of spatial transformations, including translation, rotation, scaling, shearing, and more. In [27], the affine motion model comprising 12 parameters is introduced for compressing human-shaped point clouds, with local motion estimation of each body part facilitated by the ICP algorithm. Shao et al. [17] introduce a nonrigid motion model with 16 parameters to capture 3D deformations in dynamic point clouds. Those motion model parameters would introduce an extra bitrate expense, which can be further optimized.

We highlight a selection of representative studies closely related to our study and elaborate on the advantages and disadvantages of our study compared to these works. G-PCC interEM in [21] provides a block-matching motion estimation scheme for dynamic point cloud compression. Our scheme improves upon G-PCC interEM in several ways. First, while G-PCC interEM relies solely on the previously decoded frame as the inter-reference, our forward–backward joint ME scheme enhances accuracy by generating a more suitable reference frame. Second, G-PCC interEM’s octree-based block structure limits motion representation along three axes, whereas our motion-assisted patch generation approach allows for flexible motion representation, better aligning with real motion fields. Third, while G-PCC interEM employs a translational motion model restricting motion to three DOFs, we advocate for affine motion models to increase DOFs and achieve more accurate motion approximation. However, our scheme has a disadvantage in terms of time complexity compared to G-PCC interEM due to the additional time expenses of the forward–backward joint ME scheme. P(Full) in [22] offers a context-adaptive point cloud inter coder without motion estimation, leveraging a reference octree to construct temporal contexts. While effective for most scenarios, it struggles with low-quality dynamic point cloud datasets due to difficulties in temporal context modeling caused by noise and holes. In contrast, our approach integrates forward–backward joint motion estimation, enhancing inter-geometry references and resulting in improved coding performance. Despite the higher time complexity compared to P(Full), our scheme outperforms P(Full) with significant coding gains. The Nonrigid ME in [17] offers a registration-based scheme to capture nonrigid motion in dynamic point clouds using a 16-parameter motion model. However, its iterative energy optimization process for parameter estimation results in higher computational complexity. In contrast, our method avoids iterative optimization and employs singular value decomposition for efficient affine motion computation, significantly reducing coding time. Although our approach sacrifices some motion estimation accuracy due to its simpler 12-parameter motion model compared to Nonrigid ME, it achieves faster processing times.

## 3. Our Approach

### 3.1. Overview of the Proposed Framework

The pipeline of the proposed patch-based affine ME framework is depicted in Figure 1. In the first stage, point cloud inter-frame motion analysis is conducted on previously decoded frames to capture motion prior information. Subsequently, these motion priors are utilized for frame-wise motion coherence evaluation in the second stage. Simultaneously, spatial geometry correlation among all points of the reference point cloud is evaluated using the Euclidean distance metric. Leveraging the insights gained from geometry correlation and motion prior analysis, the reference point cloud is segmented into a collection of irregularly shaped patches, each characterized by high motion consistency. Within this segmentation, each patch is linked to a control point that acts as its centroid. Consequently, the motion field of the point cloud frame is represented based on these control points. In the third stage, the forward motion field from the reference frame to the current frame is estimated as a series of control point-based affine motions via the ICP algorithm, resulting in a new affine motion-compensated reference frame. In the fourth stage, the backward ME scheme employs a block-matching method to refine motions for the current frame relative to the affine motion-compensated reference frame. Finally, the point cloud geometry and estimated motions are encoded into the total bitstream.

### 3.2. Point Cloud Inter-Geometry Motion Analysis

Prior to the ME process, forward motion tracking is conducted to capture previous motion priors. Given that the geometry information of the current frame is unavailable in the encoder, inter-geometry motion analysis is performed on previously decoded frames based on inter-frame motion consistency. The motion analysis process unfolds as follows:

Given an input point cloud sequence *F*, we arrange those frames with the group-of-picture (GOP) structure, where the first frame in a GOP is the intra frame and the following contents are inter frames. Motion analysis is conducted from the second frame in a GOP to track the motions between two consecutive frames.

Let F(t) be the decoded frame at time index *t*, it comprises *n* points $\left\{v_{1},v_{2},\ldots ,v_{n}\in R^{3}\right\}$. Let F(t−1) denote the previous frame relative to the current frame, it contains *m* points $\left\{u_{1},u_{2},\ldots ,u_{m}\in R^{3}\right\}$. The objective of motion analysis is to track motions by identifying geometric changes between consecutive frames. Motion vectors are computed by establishing correspondences between points at time *t* and points at time t−1. The nearest neighbor search method with the Euclidean distance metric is adopted to find point correspondences. Specifically, a KD-tree search is employed for efficient nearest neighbor search to mitigate computational complexity. Then, the matched point $u_{map(i)}$ in frame F(t−1) for point $v_{i}$ in frame F(t) is determined, along with the estimated motion vector $\vec{mv}\left(i\right)$ via the following optimization process:(1)$$\begin{array}{cc} \vec{mv}\left(i\right) & =argmin∥F\left(x_{i},y_{i},z_{i},t\right)-F\left(x_{i}-{mv}_{1},y_{i}-{mv}_{2},z_{i}-{mv}_{3},t-1\right)∥\\ \; & =\{\left({\hat{mv}}_{1}\left(i\right),{\hat{mv}}_{2}\left(i\right),{\hat{mv}}_{3}\left(i\right)\right)\}, \end{array}$$
(2)$$\begin{array}{cc} u_{map(i)} & =v_{i}+\vec{mv}\left(i\right)\\ \; & =(x_{i}+{\hat{mv}}_{1}\left(i\right),y_{i}+{\hat{mv}}_{2}\left(i\right),z_{i}+{\hat{mv}}_{1}\left(i\right))\\ \; & =({\hat{x}}_{j},{\hat{y}}_{j},{\hat{z}}_{j}), \end{array}$$
where $mv_{1}$, $mv_{2}$, and $mv_{3}$ are the displacements along the *x*, *y*, and *z* axis, respectively. The selection of the corresponding point $\left(x_{i}-{mv}_{1},y_{i}-{mv}_{2},z_{i}-{mv}_{3}\right)$ in the frame F(t−1) to the point $\left(x_{i},y_{i},z_{i}\right)$ in the frame F(t) involves finding the nearest neighboring point through a KD-tree search in the previous frame F(t−1). ${\hat{mv}}_{1}\left(i\right),{\hat{mv}}_{2}\left(i\right),{\hat{mv}}_{3}\left(i\right)$ are the final estimated motions. map(·) defines the point mapping operation. $\left(x_{i},y_{i},z_{i}\right)$ and $\left({\hat{x}}_{j},{\hat{y}}_{j},{\hat{z}}_{j}\right)$ are the positions of the point $v_{i}$ and $u_{map(i)}$, respectively.

Later, based on the established corresponding point pairs, the inter-frame geometry difference between two frames is evaluated by computing the square of the Euclidean L2-norm of all corresponding points. Subsequently, the total geometric changes D(F(t−1),F(t)) between the frame F(t−1) and F(t) are defined as:(3)$$D(F(t-1),F(t))=\sum_{i=1}^{n}{∥v_{i}-u_{map(i)}∥}_{2}^{2}.$$

The derived geometric changes between two consecutive frames, along with the estimated motion priors, reflect the motion characteristics in the temporal domain of the point cloud. These motion indicators play a pivotal role in subsequent motion representation and estimation processes.

### 3.3. Point Cloud Deformable Patch Generation

The conventional block-based ME technique used in point cloud compression often underperforms due to its inability to fully exploit motion correlation between adjacent blocks. This limitation can be addressed by adopting a more adaptable patch-based motion representation approach. We aim to segment point cloud inter-frames into deformable patches to better accommodate continuously varying motion fields observed in dynamic point clouds. Instead of the conventional block-based motion model, we propose a control point-based model for representing the motion field of moving point clouds. In terms of bitrate efficiency in coding motion fields, the goal of deformable patch generation is to group neighboring points with similar motions into the same patch, thereby enhancing intra-patch motion consistency. To minimize additional bitrate overhead in indicating point cloud patch generation during encoding, we perform the patch segmentation on the reference frame corresponding to the current frame to be encoded. This reference frame is decoded beforehand and is accessible to the encoder and decoder, eliminating the need for extra bitrate transmission to delineate the patch segmentation.

#### 3.3.1. A Joint Similarity Metric for Patch Generation

We propose a joint similarity metric that integrates geometry correlation and motion coherence measurement to serve as the criteria for identifying and clustering point cloud patches. Both the geometry distribution and motion characteristics within point clouds are taken into account in the generation of dynamic point cloud patches. The Euclidean distance between two points is utilized as the geometry correlation descriptor. Let $p_{i}$ and $p_{j}$ denote two adjacent points in the point cloud. The geometry correlation descriptor $geo_{-}error\left(p_{i},p_{j}\right)$ representing the geometry errors of two points in the Euclidean geometry space is defined as:(4)$$geo_{-}error\left(p_{i},p_{j}\right)={∥p_{i}-p_{j}∥}_{2}^{2},$$
where larger values of $geo_{-}error\left(p_{i},p_{j}\right)$ indicate the smaller geometric correlation between the two points, and vice versa.

Additionally, motion priors derived from the previous motion analysis stage are employed for motion field coherence detection in the point cloud, further driving the clustering of points with similar motions in a single patch. Coherent motion detection can be seen as the clustering of consistent behavior, positively impacting efficient motion field representation and bitrate savings. The motion coherence descriptor $motion_{-}error\left(p_{i},p_{j}\right)$, describing the motion discontinuity between points $p_{i}$ and $p_{j}$ in the motion field $\vec{mv}$, is defined as
(5)$$motion_{-}error\left(p_{i},p_{j}\right)={∥\vec{mv}\left(i\right)-\vec{mv}\left(j\right)∥}_{2}^{2},$$
where larger values of $motion_{-}error\left(p_{i},p_{j}\right)$ indicate smaller motion coherence between the two points, and vice versa.

Subsequently, we formulate the joint error metric $joint_{-}error\left(p_{i},p_{j}\right)$ for the similarity measure of two points $p_{i}$ and $p_{j}$, combining the geometry correlation measure $geo_{-}error\left(p_{i},p_{j}\right)$ and motion coherence measure $motion_{-}error\left(p_{i},p_{j}\right)$. The proposed joint error metric is formulated as
(6)$$\begin{array}{cc} joint_{-}error\left(p_{i},p_{j}\right)= & a_{1}\cdot geo_{-}error\left(p_{i},p_{j}\right)+a_{2}\cdot motion_{-}error\left(p_{i},p_{j}\right)\\ = & a_{1}{∥p_{i}-p_{j}∥}_{2}^{2}+a_{2}{∥\vec{mv}\left(i\right)-\vec{mv}\left(i\right)∥}_{2}^{2}, \end{array}$$
where $a_{1}$ and $a_{2}$ are two scalar parameters for the geometry correlation term $geo_{-}error\left(p_{i},p_{j}\right)$ and the motion coherence term $motion_{-}error\left(p_{i},p_{j}\right)$, respectively.

#### 3.3.2. Optimization-Based Patch Generation

With the proposed joint error metric described in Equation (Equation 6), the task of generating point cloud patches is formulated as an optimization problem, as shown in Equation (Equation 7), aiming to minimize the total joint error between the points and their respective patch centroids:(7)$$\begin{array}{cc} argmin\;total_{-}error= & \sum_{j=1}^{k}\sum_{i=1}^{g(j)}joint_{-}error\left(p_{i},\mu_{j}\right)\\ = & \sum_{j=1}^{k}\sum_{i=1}^{g(j)}a_{1}{∥p_{i}-\mu_{j}∥}_{2}^{2}+a_{2}{∥\vec{mv}\left(p_{i}\right)-\vec{mv}\left(\mu_{j}\right)∥}_{2}^{2}, \end{array}$$
where *k* is the number of patches, g(j) is the number of points in the patch $P_{j}$, and $p_{i}$ is a point within the patch $P_{j}$. The motion $\vec{mv}\left(p_{i}\right)$ of point $p_{i}$ is derived from the estimated motion field described in Equation (Equation 6). The motion $\vec{mv}\left(\mu_{j}\right)$ is approximated by the motion of the centroid’s nearest neighbor points. The centroid $\mu_{j}$ of the patch $P_{j}$ is computed as
(8)$$\mu_{j}=\frac{1}{g(j)}\sum_{i=1}^{g(j)}p_{i}.$$

The optimization-based patch generation problem is tackled through an iterative clustering process. A k-means clustering algorithm with the proposed joint error metric is devised to generate point cloud patches. Details of the proposed patch generation algorithm are presented in Algorithm 1. In the initialization phase, the desired number of point cloud patches is specified, and the patch centroids are determined using the k-means++ method [32]. In the assignment phase, the joint error between the points to be processed and the identified cluster centroids is computed using Equation (Equation 6). Points are then assigned to patches with the lowest joint error to minimize Equation (Equation 7). Later, the patch centroids are updated based on the newly determined patch assignments using Equation (Equation 8). The assignment and update operations are repeated until either the point assignment remains unchanged from the previous iteration or the preset iterations are reached.
**Algorithm 1** Proposed Patch Generation Algorithm**Input:** Point cloud with *m* points, desired number of point cloud patches *k*.**Output:** Generated point cloud patches *C* with centroids μ. 1:  Initialize patch centroids μ1,μ2,⋯,μk using the k-means++ method. 2:   iter=0, flagupdate=false. 3:  **repeat**  4:     **for** i=1:m **do**  5:         errormin(i)=0, indexmin(i)=1. 6:         **for** j=1:k **do**  7:              Compute the joint error joint−error(pi,μj) between the point pi and the patch                  centroid μj using Equation (Equation 6). 8:              **if** j==1
**or**
$error_{min}\left(i\right)>joint_{-}error\left(p_{i},\mu_{j}\right)$ **then**  9:                  $error_{min}\left(i\right)=joint_{-}error\left(p_{i},\mu_{j}\right)$. 10:                **if** $index_{min}\left(i\right)\neq j$ **then**  11:                   $index_{min}\left(i\right)=j$, $flag_{update}=\mathrm{true}$. 12:              **end if**  13:            **end if**  14:         **end for**  15:       **end for**  16:       **for** j=1:k **do**  17:           Select points belonging to the point cloud patch $C_{j}$ where $index_{min}=j$. 18:           Update the patch centroid $\mu_{j}$ with selected points using Equation (Equation 8). 19:       **end for**  20:       iter=iter+1. 21:  **until**
$iter==iter^{max}$
**or**
$flag_{update}==\mathrm{false}$ 22:  **return**
*C* and μ.


As shown in Figure 2, an example of the point cloud patch generation results is showcased from various angles. Notably, the algorithm produces intricate patches within localized regions featuring complex motions, exemplified by areas like the hair region in Redandbalck. Following the optimization-based iterative point clustering process, the point cloud is segmented into irregular patches, each containing points with homogeneous geometry distribution and motion characteristics. These point cloud patches are associated with control points acting as centroids, laying the foundation for representing the motion field of the point cloud frame. The motion field estimation for a point cloud translates into estimating the motions of all control points within point cloud patches.

### 3.4. Forward–Backward Jointing Motion Estimation

We propose a novel forward–backward jointing ME scheme tailored for approximating the motion field in dynamic point clouds. In our approach, forward ME captures motions from the reference frame to the current frame, while backward ME delineates motions from the current frame back to the reference frame. Leveraging warping in the forward ME phase enables the generation of a more refined motion-compensated reference frame, thereby enhancing the accuracy of the backward ME process. Consequently, this results in a more precise representation of the motion fields compared to conventional methods that rely solely on forward or backward ME techniques.

In the forward patch-based ME stage, we introduce affine motion models with 12 DOFs to represent the motions of point cloud patches from the reference frame to the current frame. The affine transformation of a point *p* with position (*x*, *y*, *z*) to the affine-deformed point with $p^{\prime }$ with position ($x^{\prime }$, $y^{\prime }$, $z^{\prime }$) can be defined as
(9)$$\left[\begin{array}{c} p^{\prime }\\ 1 \end{array}\right]=\left[\begin{array}{cc} R & T\\ 0 & 1 \end{array}\right]\left[\begin{array}{c} p\\ 1 \end{array}\right],$$
(10)$$\left[\begin{array}{c} x^{\prime }\\ y^{\prime }\\ z^{\prime }\\ 1 \end{array}\right]=\left[\begin{array}{cccc} r_{11} & r_{12} & r_{13} & t_{1}\\ r_{21} & r_{22} & r_{23} & t_{2}\\ r_{31} & r_{32} & r_{33} & t_{3}\\ 0 & 0 & 0 & 1 \end{array}\right]\left[\begin{array}{c} x\\ y\\ z\\ 1 \end{array}\right],$$
where the matrix *R* with parameter $r_{ij}$ represents a 3 × 3 affine transformation matrix with 9 DOFs. It is crucial to emphasize that this matrix accounts for various affine motions within dynamic point clouds, including zooming, rotation, scaling, and shear mapping, each contributing to different DOFs. Since scaling and shearing are not rigid transformations, the affine transformation matrix *R* with 9 DOFs is not rigid. *T* with parameter $t_{i}$ denotes translation along different directions.

Every control point identified during the previous patch generation process is linked with an affine transformation detailing its transition from the preceding frame to the current frame. These transformations are determined by locating the optimal matching point in the current frame for each control point in the previous frame. We utilize the classic ICP algorithm [33] to construct the point correspondences. This is achieved by identifying the nearest neighbor point pairs between the point cloud patch centered at the control point in the reference frame and points within a search window surrounding the control point’s position in the current frame.

With those established point correspondences between the point $u_{i}\in P$ in the reference frame and the match point $v_{map(i)}\in P_{map}$ in the current frame, the affine transformation is obtained by solving the following optimization problem:(11)$$argmin\;\sum_{u_{i}\in P}{∥v_{map(i)}-\left(Ru_{i}+t\right)∥}_{2}^{2},$$
where *P* denotes the patch to be processed in the reference frame, and $P_{map}$ means the matched point set in the current frame.

We employ singular value decomposition (SVD) to solve the optimization problem in Equation (Equation 11). First, we compute the weighted centroids $\bar{v}$ and $\bar{u}$ for the point sets *P* and $P_{map}$, each with *n* and *m* points, respectively, as $\bar{v}=\frac{1}{n}\sum_{i=1}^{n}v_{i}$ and $\bar{u}=\frac{i=1}{m}\sum_{1}^{m}u_{i}$. Next, we derive the mean-centered point sets $\tilde{P}$ and $\tilde{P}map$ by subtracting the mean from *P* and Pmap. We compute the covariance matrix *S* between $\tilde{P}$ and $\tilde{P}map$ as $S=\tilde{P}W\tilde{P}{map}^{T}$, where *W* is a diagonal matrix. Then, we decompose the matrix *S* using SVD as:(12)$$S=U\Sigma V^{T},$$
where *U* is a left singular vector matrix, $\Sigma =diag(\sigma_{i})$ is a diagonal matrix containing singular values $\sigma_{i}$, and *V* is a right singular vector matrix.

Later, the affine transformation matrix *R* and the translation vector *t* describing the affine motion from the reference centered point set $\tilde{P}$ to the current centered point set ${\tilde{P}}_{map}$ can be defined as:(13)$$R=V\Sigma^{*}U^{T},$$
(14)$$t={\tilde{P}}_{map}-R\tilde{P},$$
where the matrix $\Sigma^{*}$ is a diagonal matrix with elements $1/\sigma_{i}$, if $\sigma_{i}$ greater than zero, and zero otherwise. The rotation action in the affine transformation matrix *R* is constructed from the matrix *V* and the matrix *U*. The scaling factors, found in the diagonal elements of $\Sigma^{*}$, govern scaling along different axes, enabling both uniform and non-uniform zooming effects. Additionally, the shear operation, integrating aspects of scaling with rotations, further contributes to the comprehensive representation of affine motions.

The resulting affine transformation matrices for all patches are encoded and transmitted to facilitate motion estimation on the decoded point clouds. Utilizing the obtained patch-wise affine motion field, affine motion compensation occurs via an affine transformation, mapping point cloud patches from the reference frame to the current frame. This process yields a newly wrapped reference frame with a smooth reconstruction.

In the backward block-based ME stage, compensated frames from the forward ME serve as inter references. A backward ME approach is then employed to capture local motions from the current frame to the reference frame, enhancing the motion estimation accuracy. Utilizing the block-matching ME method, as in interEM [21], the current point cloud is initially partitioned into blocks of varying sizes, supported at dimensions of 32 × 32 × 32 and 16 × 16 × 16. A three-step search motion algorithm, detailed in [17], identifies optimal translational motions for occupied blocks. Subsequently, these refined motions are encoded into the bitstream utilizing the arithmetic coder integrated within the interEM framework [21], with tailored contexts designed to enhance compression efficiency.

## 4. Experimental Results

We perform a series of experiments to evaluate the effectiveness and efficiency of the proposed R-D optimized ME framework for dynamic point cloud geometry compression. Details of our experimental setup are provided in Section 4.1. The comparison between the proposed scheme with competitive platforms in terms of compression performance and time complexity is presented in Section 4.2 and Section 4.3, respectively. Moreover, ablation studies are conducted in Section 4.4 to validate the performance of our key processing modules.

### 4.1. Simulation Setup

All experiments are conducted using a computer (Lenovo, Beijing, China) equipped with an Intel i7 8700K CPU (3.7 GHz) and 64 GB RAM. We employ a selection of standard point cloud sequences with diverse motion characteristics sourced from MPEG [34] and JPEG [35] as our test datasets. The visualization of these datasets is provided in Figure 3. The first 200 frames of each sequence are tested for coding performance evaluation. The characteristics of those datasets are provided in Table 1. G-PCC interEM [21] serves as the inter codec anchor for dynamic point cloud geometry compression within our framework. Our proposed approach is compared against several competitive platforms from both industry and academia: (i) **G-PCC interEM w/ME** (enabled motion estimation), (ii) **G-PCC interEM w/o ME** (disabled motion estimation), (iii) **Nonrigid ME** (nonrigid registration-based motion estimation scheme [17], representing our previous work), and (iv) **P(Full)** (context-based inter codec [22]). Adhering to MPEG common test conditions [34], we use bits per point (bpp) to denote the total geometry bitrate. The geometry bitrate gain, calculated as $\left(1-bitrate_{a}/bitrate_{b}\right)$, illustrates the coding performance improvement of method *a* with bitrate $bitrate_{a}$ over method *b* with bitrate $bitrate_{b}$. To ensure a fair comparison across all tests, the GOP is uniformly defined as 8.

### 4.2. Compression Performance Evaluation

Table 2 provides a comprehensive summary of the compression performance evaluation of the proposed scheme against competitive platforms in dynamic point cloud geometry lossless compression. Specifically, the proposed scheme achieves significant geometry bitrate gains, averaging 6.28%, 7.33%, 1.77%, and 16.57% when compared to interEM w/ ME, interEM w/o ME, Nonrigid ME, and P(Full), respectively. These outcomes consistently demonstrate the superior coding efficiency of our approach for dynamic point cloud geometry compression. The comparative experimental results in Table 2 underscore the importance of a meticulously crafted ME scheme. Specifically, the P(Full) approach, acting as a context-based inter codec that omits ME in dynamic point cloud geometry compression, demonstrates notably inferior coding performance across various sequences, particularly in JPEG sequences characterized by simpler point cloud motions. In contrast, our proposed ME scheme, featuring a novel motion representation and estimation approach, excels in capturing precise forward–backward jointing motion fields with minimal overhead, thereby enhancing both the accuracy of inter-frame context and overall inter coding performance. This enhancement is evidenced by an average geometry bitrate gain of 16.57% over P(Full). Conversely, P(Full) without ME struggles to achieve comparable accuracy and efficiency in compressing different point cloud sequences.

Moreover, the experiments highlight the efficacy of the proposed affine ME scheme in enhancing the motion model’s capability to capture intricate motions within point cloud sequences. As illustrated in Table 2, our ME scheme consistently outperforms the Nonrigid ME scheme across all tests, achieving an average 1.77% geometry bitrate gain. The Nonrigid ME scheme is our previous work on dynamic point cloud compression, which introduces a nonrigid motion model with 16 parameters to capture point cloud complex motions. Our proposed affine ME scheme devises an affine motion model with an enhanced patch-based motion representation that can better depict the motion field in dynamic point clouds. With only 12 model parameters, our affine motion model requires four fewer parameters to be encoded into the bitstream compared to the Nonrigid ME scheme. Moreover, Figure 4 presents the frame-wise compression performance of the proposed scheme and comparative platforms. Substantial and consistent geometry bitrate gains are obtained by the proposed scheme across all tests, which validate the effectiveness and robustness of the proposed scheme for compressing dynamic point clouds with diverse motion features. Moreover, there are some peaks in geometry bitrate in Figure 4, which are attributed to the GOP parameter. We adopt the IPPP coding structure with GOP = 8, where the first frame in a GOP serves as the I frame, while the subsequent frames in the GOP are P frames that utilize the previously decoded frame as the inter reference for inter prediction and coding. Generally, the geometry bitrate in I frames tends to be larger than that in P frames, resulting in the observed geometry bitrate peaks in Figure 4.

To further evaluate the coding ability of the proposed framework in handling dynamic point clouds under diverse conditions, we conduct a series of experiments to test our scheme under different frame rates on two datasets characterized by distinct noise levels. The experiments involve performing motion estimation consecutively, then on every 5th frame, and subsequently on every 10th, 15th, and 30th frame. These frame rates are chosen to represent various levels of temporal granularity in the motion estimation process. We utilize two representative datasets for evaluation: Phil, which represents a low-quality sparse point cloud with significant noise, and Longdress, which represents a high-quality dense point cloud. By conducting experiments on these datasets under different frame rates, we aim to assess the robustness and effectiveness of our scheme in various scenarios encountered in practical applications of dynamic point cloud compression. Table 3 summarizes the compression performance of our proposed scheme under different frame rate settings for dynamic point cloud geometry lossless compression. We use the total geometry bitrate to denote coding performance and geometry PSNR values to represent the reconstruction qualities of motion-compensated point clouds. The experimental results demonstrate the consistent compression performance achieved by our scheme across point cloud sequences with varying frame rates. As the interval between frames increases, leading to larger motion between frames, we observe a marginal increase in the geometry bitrate. Despite this, our scheme effectively handles the heightened motion between consecutive frames, ensuring high-quality motion-compensated point clouds. These findings verify the robustness and adaptability of our proposed scheme in handling dynamic point cloud sequences with varying frame rates.

Additionally, Figure 5 and Figure 6 present the reconstruction quality of motion-compensated point clouds generated by our scheme at different frame rate settings on Phil and Longdress, respectively. Each row within the figures showcases original point clouds at two frame indices alongside motion-compensated point clouds from three different perspectives (front, left, and back). We also visualize the mean square error of the geometry distortion between the original and motion-compensated point clouds with error color-coding maps. The experimental results verify the ability of our scheme to generate high-quality motion-compensated point clouds across diverse datasets characterized by distinct noise levels. In our paper, the GOP parameter is not manually tuned but empirically set, according to the commonly adopted GOP = 8 in video coding tests. It is crucial to highlight that our scheme utilizes the IPPP coding structure without B frames. Consequently, variations in the GOP settings do not exert a significant influence on the motion estimation performance of our scheme. Moreover, our proposed forward-backward joint motion estimation approach distinguishes itself from bidirectional prediction methods. Unlike bidirectional prediction methods that necessitate referencing both the previous and subsequent frames, our scheme exclusively utilizes the previous frame as a reference. This design enables our scheme to consistently deliver optimal performance across different GOP settings, thereby ensuring robustness and stability in various coding scenarios.

### 4.3. Compression Complexity Evaluation

Table 4 and Table 5 present the second-frame and 32-frame runtime results of the proposed scheme compared to competitive platforms in dynamic point cloud geometry lossless compression, respectively. Analyzing the coding times of both the second frame and the entire 32 frames in tested point cloud sequences enables a comprehensive complexity evaluation of different compression methods. This evaluation assesses both short-term performance and long-term stability. With the coding performance results presented in Table 2 and the runtime results presented in Table 4 and Table 5, we can conduct a comprehensive performance analysis of our approach and the competitive platforms.

Compared to G-PCC interEM, the proposed scheme exhibits an increased coding time of 18% and 24.64% in the second-frame and 32-frame tests, respectively. These additional time expenses are attributed to the proposed forward–backward joint motion estimation scheme. It introduces additional computational overhead compared to the single-direction motion estimation employed by G-PCC interEM. Despite the increased coding time, our approach consistently achieves 6.28% geometry bitrate gains over G-PCC interEM, validating the effectiveness of our forward–backward joint motion estimation scheme. In contrast to Nonrigid ME, our scheme significantly reduces coding time by 99.65% and 99.64% in the second-frame and 32-frame tests, respectively. This runtime reduction stems from our decision to avoid the iterative optimization-based motion estimation utilized in Nonrigid ME. Instead, we employ singular value decomposition for efficient and accurate affine motion computation. As the source code of the platform P(Full) is unavailable, we use the runtime results provided in the paper [22] for comparison. However, it is important to note that only partial second-frame runtime results of P(Full) for compressing certain representative sequences are available in the paper. In contrast to P(Full), which is a context-based inter codec without motion estimation, our scheme typically incurs additional coding time due to the introduction of the proposed motion estimation process in most tests. Nonetheless, our approach consistently outperforms P(Full) with an average geometry bitrate gain of 16.57%, confirming the effectiveness of our motion estimation scheme. Notably, in the runtime test on Soldier, our scheme demonstrates a 3.10% coding time reduction compared to P(Full). This reduction may be attributed to the time-consuming spatial–temporal context construction process in P(Full), whereas our motion estimation process proves to be more efficient.

Combining the experimental results on coding performance and time complexity, these outcomes consistently demonstrate that, compared to G-PCC interEM, our approach can achieve significant coding gains with an acceptable runtime increase. Moreover, in comparison to Nonrigid ME, our approach not only enhances coding performance but also significantly reduces coding time for dynamic point cloud geometry compression. Moreover, when compared to P(Full), although such inter codecs without motion estimation are effective in saving encoding time, the coding performance of our scheme surpasses them by a significant margin.

### 4.4. Ablation Studies

We perform ablation studies to objectively verify the effectiveness of key modules in the proposed framework. Validation results of all studies are presented in Table 6. We independently analyze each module as follows.

**Validation of the Forward–Backward Jointing ME Strategy:** Our proposed forward–backward jointing ME scheme addresses the challenge of inaccurate block-matching pairs encountered in the backward ME process, as discussed in Section 3. These inaccuracies can lead to imprecise motion estimates, consequently impacting inter coding performance negatively. In our approach, forward ME captures motions from the reference frame to the current frame, while backward ME delineates motions from the current frame back to the reference frame. Leveraging warping in the forward ME phase facilitates the generation of a more refined motion-compensated reference frame, thereby improving the accuracy of the backward ME process. Table 6 provides empirical evidence supporting the effectiveness of our joint ME strategy, demonstrating an average geometry bitrate gain of 2.64% in our proposed framework. This validation underscores how our ME scheme enhances the accuracy of geometry matching in the motion search process for dynamic point cloud geometry compression. By exploiting warping in the forward ME phase, we achieve a more precise representation of motion fields, surpassing conventional methods reliant solely on forward or backward ME techniques.

**Validation of the Patch-based Motion Field Representation:** Given the inherent limitations of the octree-based block structure in the interEM, which restricts motion representation to capturing discontinuities along fixed directions, we introduce a motion-assisted patch generation approach. Our method enables flexible motion representation, facilitating improved alignment between the estimated motion representation and the actual motion field in point clouds. Table 6 demonstrates that our proposed patch-based motion representation model achieves an average geometry bitrate gain of 1.94% compared to the block-based motion model in the interEM. These results affirm the advantages of our proposed patch-based motion field representation, illustrating how our irregular patch representation can more accurately capture the diverse local motions present in point clouds. Furthermore, Figure 7 showcases the deformable patch generation results with control point representation across all test datasets, reinforcing the advantages and robustness of our proposed patch-based motion field representation across various point clouds exhibiting diverse motion features.

**Validation of the Affine Motion Model:** Recognizing the limitations of the translational motion model in the interEM, which hampers its ability to capture complex motions in point clouds, we advocate for integrating affine motion models to enable more accurate motion approximation with increased DOFs. To assess the effectiveness of our enhanced affine motion model for compressing point cloud sequences, we conduct a comparative test between the proposed affine motion model and the translational model used in the interEM. Analysis of the experimental results presented in Table 6 reveals significant geometry bitrate gains achieved by the proposed affine motion model, averaging 2.53% on all test datasets. Introducing the affine motion model equips the encoder with the capability to precisely characterize rotation, zooming, and object deformation, thereby facilitating more effective motion-compensated prediction for efficient point cloud compression. These experimental findings validate the efficacy of the proposed affine motion model within our framework for dynamic point cloud geometry compression.

To further verify the effectiveness of our proposed motion model, we conduct a detailed affine transformation analysis by visualizing selected affine transformations estimated by our scheme. As depicted in Figure 8, we select two representative patches from the original point cloud: the 4th patch representing the hair with complex geometric details and the 183rd patch representing the foot with intricate local deformations. Our proposed scheme efficiently estimates affine transformations with 12 parameters describing local motions between the original and target point clouds. The resulting affine-transformed local patches aligned excellently with the target point cloud, confirming the effectiveness of our scheme in capturing affine motions in dynamic point clouds.

## 5. Conclusions

In this paper, we propose a novel patch-based affine ME framework for dynamic point cloud geometry compression. We propose a forward–backward jointing ME strategy, employing affine motion-compensated frames to enhance inter-geometry references. Leveraging the proposed ME strategy, we achieve improved ME accuracy by generating motion-compensated frames and refining motions iteratively. Moreover, by conducting motion analysis and segmenting point clouds into deformable patches, our motion-assisted patch generation scheme enables flexible representation of point cloud motions, enhancing compression efficiency. Furthermore, we introduce an affine motion model to replace the traditional translational model to improve the ME accuracy for dynamic point clouds. These advancements offer promising solutions for efficient dynamic point cloud compression in various applications. Experimental results demonstrate that the proposed framework surpasses various competitive platforms in terms of compression performance. Moreover, ablation studies can prove the effectiveness of key modules in our scheme.

## Figures and Tables

**Figure 1 sensors-24-03142-f001:**
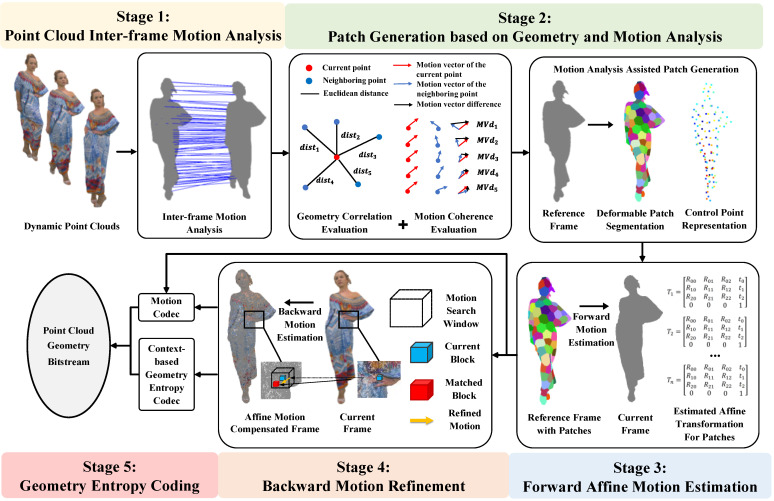
The pipeline of the proposed patch-based affine ME scheme for dynamic point cloud geometry compression.

**Figure 2 sensors-24-03142-f002:**
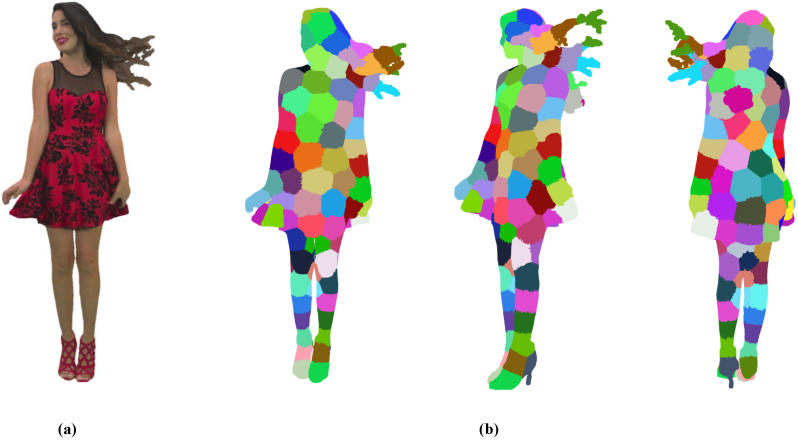
Demonstration of point cloud patch generation from various angles for dataset Redandblack: (**a**) Original point cloud. (**b**) Point cloud generation results presented from different Angles.

**Figure 3 sensors-24-03142-f003:**
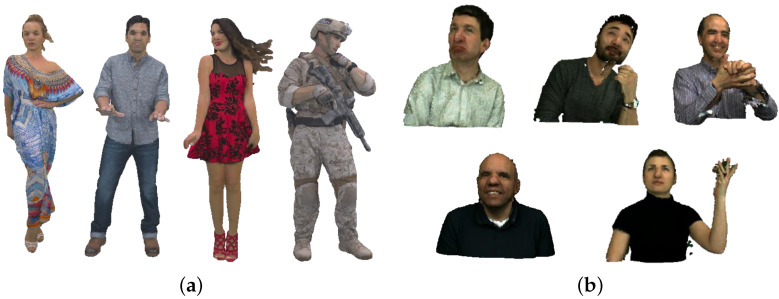
Point cloud datasets. (**a**) MPEG 3DG point clouds. Their first frames are shown from left to right: Longdress, Loot, Redandblack, and Soldier. (**b**) JPEG Pleno point clouds. Their first frames are shown from left to right: Andrew, David, Phil, Ricardo, and Sarah.

**Figure 4 sensors-24-03142-f004:**
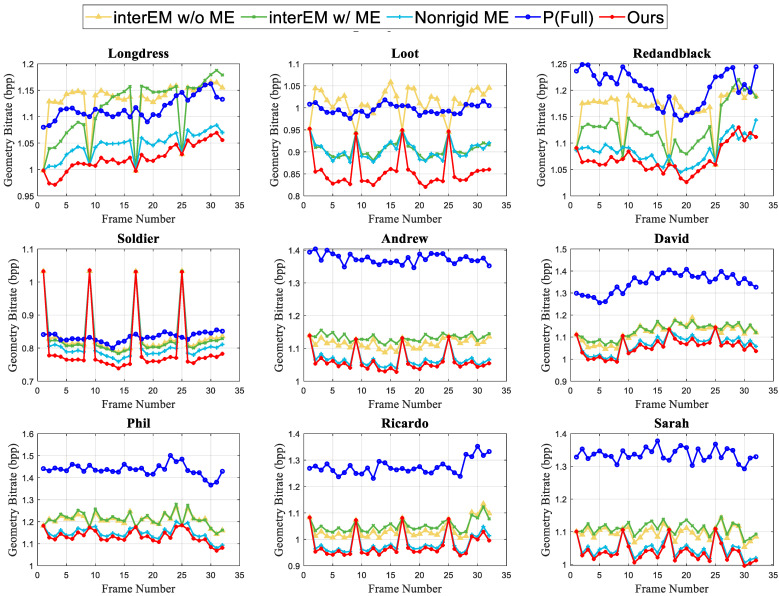
Frame-wise compression performance comparison between the proposed scheme and the state-of-the-arts on test datasets. The first 32 frames of each sequence with GOP = 8 are tested.

**Figure 5 sensors-24-03142-f005:**
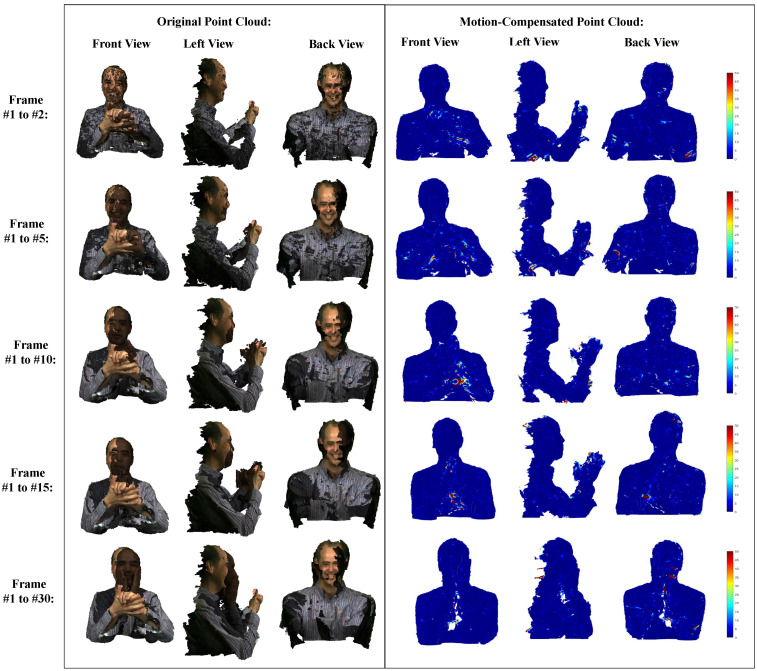
The reconstruction quality of motion-compensated point clouds generated by the proposed scheme at different frame rate settings on the dataset Phil.

**Figure 6 sensors-24-03142-f006:**
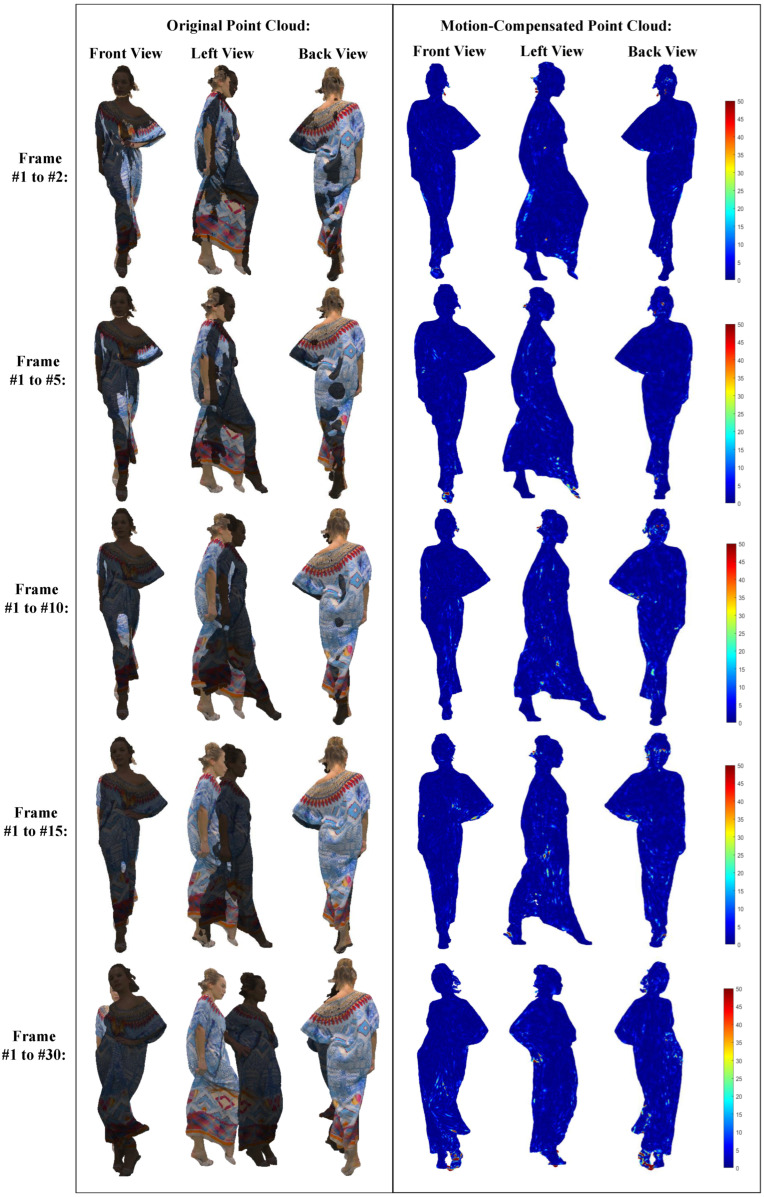
The reconstruction quality of motion-compensated point clouds generated by the proposed scheme at different frame rate settings on the dataset Longdress.

**Figure 7 sensors-24-03142-f007:**
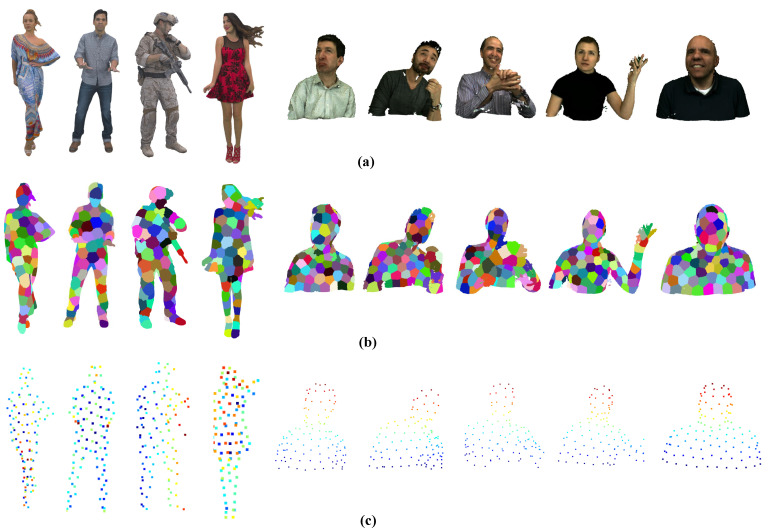
Point cloud patch generation results with the control point representation for all test datasets. (**a**) Original point cloud. (**b**) Point cloud patch generation. (**c**) Control point representation.

**Figure 8 sensors-24-03142-f008:**
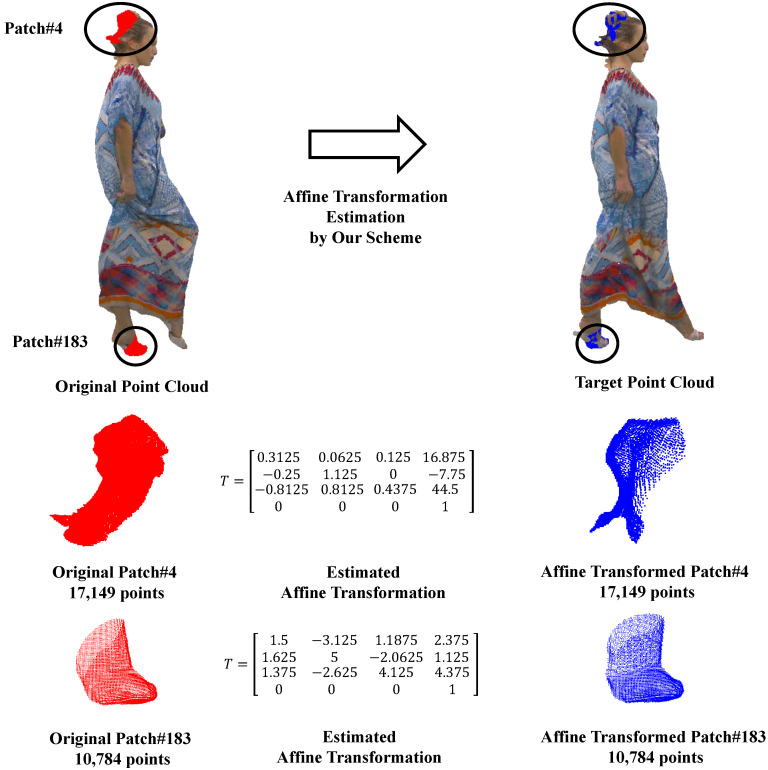
An example of the affine transformation estimation by the proposed scheme on Longdress.

**Table 1 sensors-24-03142-t001:** Characteristics of point cloud datasets.

Category	Sequence	Geometry Precision	Test Frame Range	Total Point Number
MPEG	Longdress	10	1051∼1082	26,128,187
	Loot	10	1000∼1031	25,741,340
	Redandblack	10	1450∼1481	23,392,394
	Soldier	10	536∼567	35,008,852
JPEG	Andrew	9	0∼31	9,314,433
	David	9	0∼31	10,612,744
	Phil	9	0∼31	11,532,439
	Ricardo	9	0∼31	7,047,590
	Sarah	9	0∼31	9,984,193

**Table 2 sensors-24-03142-t002:** Geometry lossless compression performance comparison between the proposed scheme and the states of the art.

Category	Sequence	Geometry Bitrate (bpp)					Geometry Bitrate Gain of Ours			
		Ours	interEM w/ME	interEM w/o ME	Nonrigid ME	P(Full)	interEM w/ME	interEM w/o ME	Nonrigid ME	P(Full)
MPEG	Longdress	1.0217	1.1144	1.1272	1.0456	1.1163	8.32%	9.36%	2.28%	8.47%
	Loot	0.8555	0.9065	1.0118	0.9049	0.9971	5.62%	15.45%	5.46%	14.20%
	Redandblack	1.0703	1.1307	1.1649	1.0854	1.2059	5.34%	8.12%	1.39%	11.24%
	Soldier	0.7980	0.8365	0.8411	0.8106	0.8333	4.60%	5.12%	1.55%	4.23%
JPEG	Andrew	1.0594	1.1328	1.1161	1.0681	1.3738	6.48%	5.08%	0.81%	22.88%
	David	1.0572	1.1277	1.1195	1.0701	1.3471	6.25%	5.57%	1.21%	21.52%
	Phil	1.1333	1.2119	1.2076	1.1484	1.4359	6.48%	6.15%	1.31%	21.07%
	Ricardo	0.9771	1.0514	1.0372	0.9868	1.2740	7.07%	5.79%	0.98%	23.30%
	Sarah	1.0392	1.1098	1.0975	1.0488	1.3359	6.36%	5.31%	0.92%	22.21%
**Average Results**		1.0013	1.0691	1.0803	1.0187	1.2133	**6.28%**	**7.33%**	**1.77%**	**16.57%**

**Table 3 sensors-24-03142-t003:** The compression performance of the proposed scheme under different frame rate settings.

Category	Sequence	Coding Metric	Frame 1 to 2	Frame 1 to 5	Frame 1 to 10	Frame 1 to 15	Frame 1 to 30
MPEG	Longdress	Bitrate (bpp)	1.01	1.03	1.04	1.05	1.08
		PSNR (dB)	64.13	61.65	65.64	64.78	64.33
JPEG	Phil	Bitrate (bpp)	1.14	1.14	1.18	1.13	1.10
		PSNR (dB)	56.89	52.17	57.35	55.76	56.34

**Table 4 sensors-24-03142-t004:** The second-frame point cloud geometry lossless compression runtime comparison between the proposed scheme and the states of the art.

Category	Sequence	Geometry Coding Time (s)				Time Reduction of Ours		
		Ours	G-PCC	Nonrigid	P(Full)	G-PCC	Nonrigid	P(Full)
			interEM	ME		interEM	ME	
MPEG	Longdress	102.03	76.20	238.95	–	−33.89%	99.83%	–
	Loot	82.17	68.02	128.25	37.40	−20.81%	99.70%	−242.91%
	Redandblack	79.58	66.64	212.17	36.99	−19.41%	99.84%	−473.59%
	Soldier	54.33	46.98	56.28	58.08	−15.63%	99.03%	3.10%
JPEG	Andrew	19.86	16.30	42.22	–	−21.86%	99.67%	–
	David	24.27	20.13	36.22	–	−20.57%	99.57%	–
	Phil	36.09	30.67	80.55	14.73	−17.68%	99.79%	−446.82%
	Ricardo	11.44	10.28	30.88	–	−11.24%	99.70%	–
	Sarah	21.30	19.58	56.11	–	−8.78%	99.72%	–
**Average Results**		47.90	39.42	97.96	**–**	**−18.88%**	**99.65%**	**–**

**Table 5 sensors-24-03142-t005:** The 32-frame point cloud geometry lossless compression runtime comparison between the proposed scheme and the States of the art.

Category	Sequence	Geometry Coding Time (s)			Time Reduction of Ours	
		Ours	G-PCC	Nonrigid	G-PCC	Nonrigid
			interEM	ME	interEM	ME
MPEG	Longdress	2533.64	1983.41	7837.63	−27.74%	99.85%
	Loot	2308.88	1757.34	3504.73	−31.38%	99.65%
	Redandblack	2318.56	1777.13	7148.50	−30.47%	99.85%
	Soldier	1477.08	1263.98	1589.06	−16.86%	98.95%
JPEG	Andrew	604.49	474.00	947.40	−27.53%	99.56%
	David	782.61	585.41	1068.38	−33.69%	99.57%
	Phil	978.38	777.80	3441.86	−25.79%	99.85%
	Ricardo	366.92	318.10	1412.54	−15.35%	99.78%
	Sarah	591.27	523.39	1701.01	−12.97%	99.75%
**Average Results**		1329.09	1051.17	3183.46	**−24.64%**	**99.64%**

**Table 6 sensors-24-03142-t006:** Ablation studies on the key modules in the proposed affine ME framework.

Category	Sequence	Geometry Bitrate (bpp)				Geometry Bitrate Gain of Ours		
		Ours	w/o	w/o	w/o	w/o	w/o	w/o
			Joint ME	Patch	Affine	Joint ME	Patch	Affine
MPEG	Longdress	1.0217	1.0775	1.0610	1.0736	5.18%	3.70%	4.83%
	Loot	0.8555	0.8830	0.8651	0.8783	3.11%	1.11%	2.59%
	Redandblack	1.0703	1.0996	1.0898	1.0959	2.66%	1.79%	2.33%
	Soldier	0.7980	0.8129	0.8000	0.8084	1.84%	0.25%	1.28%
JPEG	Andrew	1.0594	1.0809	1.0818	1.0835	1.99%	2.07%	2.22%
	David	1.0572	1.0825	1.0799	1.0829	2.34%	2.11%	2.38%
	Phil	1.1333	1.1668	1.1616	1.1655	2.87%	2.43%	2.76%
	Ricardo	0.9771	0.9941	0.9972	0.9983	1.71%	2.02%	2.12%
	Sarah	1.0392	1.0615	1.0602	1.0626	2.10%	1.98%	2.21%
**Average Results**		1.0013	1.0288	1.0218	1.0277	**2.64%**	**1.94%**	**2.53%**

## Data Availability

The original contributions presented in the study are included in the article, further inquiries can be directed to the first author.

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
