# Peer review of "Advanced Patch-Based Affine Motion Estimation for Dynamic Point Cloud Geometry Compression"

_sensors, 2024, doi:10.3390/s24103142_

Round 1
Reviewer 1 Report
Comments and Suggestions for Authors
Please see my review in the pdf file attached.

Author Response
Dear Reviewer,
We sincerely appreciate your time and effort in providing valuable feedback and comments on our manuscript titled "Advanced Patch-based Affine Motion Estimation for Dynamic Point Cloud Geometry Compression." Your insightful suggestions have been significant in improving the quality and clarity of our work.
We have diligently addressed each of your comments and suggestions in the revised manuscript. Specifically, we have incorporated revisions highlighted in blue text throughout the document, ensuring that all concerns have been thoroughly addressed.
Please see the attachment which includes a cover letter, a detailed revision report outlining the changes made in response to your feedback, and our revised manuscript.
Thanks very much for your time and efforts in handling our revised manuscript!
Regards,
Authors

Reviewer 2 Report
Comments and Suggestions for Authors
The authors propose an alternative solution to the existing motion estimation concept in geometry-based point cloud compression. In this context, the authors explore the application of bidirectional motion-estimation schemes and motion-assisted patch generation to improve fine-tuning of the motion. Additionally, they also advocate the incorporation of an affine motion model to support any warping operations. The paper provides a significant novel contribution. The flow of the writing is commendable. However, a few aspects of the paper need improvement. My comments are as follows.
1. Revisit equation (1) which involves two frames F(t) with n points and F(t-1) with m points. The previous/reference frame may not represent a point (xi-mv1). Therefore, I suppose the neighbouring point is approximated by KD-tree search. If so, it is necessary to mention the KD-tree search operation in the equation.
2. Revisit the equation(5). It is written as mv(i) – mv(i).
3. Rewrite the Optimization-based Patch Generation section with the correct use of parameters. For example, there is no l in equation (7). Revisit equation (8) as well.
4. Use parametric representation to denote a patch in equations and descriptions.
5. The misuse of notations confuses a lot. Line 191 reads “the matched point umap(i) in frame F(t − 1) for point vi in frame F(t) is determined” whereas line 303 reads “With those established point correspondences between the point vi ∈ P in the reference frame and the match point umap(i) ∈ Pmap in the current frame”. Does vi refer to a point current frame or past frame?
6. Discuss the advantages and disadvantages of your methodology with some of the works reported in the related work section.
7. Discuss the complexity of your approach.
8. It would be interesting to visually display some of the selected affine transformations.
Author Response

(The authors gave the same response as above.)

Round 2
Reviewer 2 Report
Comments and Suggestions for Authors
The comments have been duly addressed.